# Design of Ultra-Compact Optical Memristive Switches with GST as the Active Material

**DOI:** 10.3390/mi10070453

**Published:** 2019-07-05

**Authors:** Ningning Wang, Hanyu Zhang, Linjie Zhou, Liangjun Lu, Jianping Chen, B.M.A. Rahman

**Affiliations:** 1State Key Laboratory of Advanced Optical Communication Systems and Networks, Department of Electronic Engineering, Shanghai Jiao Tong University, Shanghai 200240, China; 2Department of Electrical and Electronic Engineering, City, University of London, London EC1V 0HB, UK

**Keywords:** optical switch, phase change material, integrated silicon photonic circuits, nanophononics

## Abstract

In the following study, we propose optical memristive switches consisting of a silicon waveguide integrated with phase-change material Ge_2_Sb_2_Te_5_ (GST). Thanks to its high refractive index contrast between the crystalline and amorphous states, a miniature-size GST material can offer a high switching extinction ratio. We optimize the device design by using finite-difference-time-domain (FDTD) simulations. A device with a length of 4.7 μm including silicon waveguide tapers exhibits a high extinction ratio of 33.1 dB and a low insertion loss of 0.48 dB around the 1550 nm wavelength. The operation bandwidth of the device is around 60 nm.

## 1. Introduction

Integrated chip-level photonic circuits play a pivotal role in optical communication systems and networks [1]. Active devices such as switches and modulators of high performances are highly demanded. The optical switch, as one of the most fundamental components, has got widely studied. Thermo-optic (TO) switches has been successfully demonstrated on the silicon-on-insulator (SOI) platform [1,2], which incorporates hundreds of passive optical components and active tuners. The TO effect is relatively slow, limiting the switching speed in the order of microsecond. On the other hand, the electro-optic (EO) switches based on the free-carrier plasma dispersion effect can have a much higher speed up to nanosecond. High-radix EO switches have also been realized [3,4]. Although the basic switching elements and the overall switching topology can be further optimized to improve the switching performances, there are fundamental limitations for the TO and EO switches. Firstly, the switching state can only be maintained when there is a sustainable external power supply. This leads to large static power consumption. Second, as the refractive index tuning efficiency in both TO and EO methods is relatively low, it requires a long active waveguide of at least 10’s micrometers to make the switch, which is disadvantageous for high-density photonic integration. 

Phase change materials (PCMs), which have an extra-high complex refractive index contrast between two phase states, can be used to mitigate the above issues [5,6,7]. In recent years, the combination of nanophotonic components and PCM has been studied intensively. For example, vanadium dioxide (VO_2_) with semiconductor and metal phases [7] can be used to make optical switches. However, as VO_2_ is a volatile PCM, it is difficult to maintain the metal phase without power consumption. In contrast, another common PCM, Ge_2_Sb_2_Te_5_ (GST), has the “self-holding” feature, which means it does not need a continuous power supply to keep its state. Besides, the phase transition of GST occurs on a sub-nanosecond time scale [5]. Therefore, GST is a good material choice for implementing non-volatile miniature photonic devices [8,9]. 

Here, we propose ultra-compact optical memristive switches that exploit GST as the active material connecting two tapered silicon waveguides. The optical memristive switches can provide two or multiple distinct optical transmission states based on resistive switching with a memory effect, i.e., the switching is non-volatile [10,11,12,13,14]. Once the state of the switch is triggered, it can be kept with no power consumption. The silicon and GST waveguides are designed to have matched effective indices and modal profiles to minimize the mode transition loss. The switch with a 0.7-μm-long GST section exhibits an insertion loss of <0.5 dB and a high extinction ratio of >30 dB. The Si-GST hybrid platform enables dense integration of high-performance switching components, opening new avenues for future low-power non-volatile photonic integrated circuits. 

## 2. Device Structure and Analysis

The phase change material, GST, has two phases: amorphous phase and crystalline phase. Figure 1a shows the refractive indices (both real and imaginary parts) between the two phases. The refractive index information is measured from a thin GST film by spectroscopic ellipsometry. The phase change can be induced thermally, optically, or electrically potentially with an ultra-high speed [15,16,17,18]. Additionally, both of two phases have the “self-holding” feature to maintain its state at room-temperature without continuous power supply. Figure 1b illustrates the device structure composed of a silicon waveguide inserted with a section of GST. When the GST is at the amorphous phase, light can be transmitted through with a high output power (ON-state). However, when GST becomes crystalline, light is highly absorbed by GST, so the output power is very low (OFF-state). In order to build a high-performance on-off switch, the device insertion loss (IL) of the ON-state should be low, and the output power extinction ratio (ER) between the ON-state and the OFF-state should be large.

The device insertion loss contains two parts, the GST absorption loss and the coupling loss between the silicon waveguide and the GST section. Both of these losses should be taken into consideration when we optimize the device. We first analyze the GST waveguide without considering its connection with the silicon waveguide. The effective complex refractive indices of GST waveguide are expressed as n_eff,am_ = n_1_ + *i*k_1_ and n_eff,cr_ = n_2_ + *i*k_2_ for its amorphous and crystalline phases, respectively. The optical transmission loss is determined by the attenuation coefficient, k_1,2_. The transmissivity through the GST section, T, is given by
(1)$$T=e^{\frac{-4\pi }{\lambda }k_{1,2}L}$$
the optical transmission loss at the two states can be written as,
(2)$${\mathrm{loss}}_{\mathrm{am}}=-10\log_{10}\cdot T_{\mathrm{am}}=\frac{40\pi }{\lambda }k_{1}L\cdot \log_{10}e$$
(3)$${\mathrm{loss}}_{\mathrm{cr}}=-10\log_{10}\cdot T_{\mathrm{cr}}=\frac{40\pi }{\lambda }k_{2}L\cdot \log_{10}e$$
the propagation extinction ratio (PER) of the GST waveguide between the ON-state and the OFF-state is written as:(4)$$\mathrm{PER}=\frac{40\pi }{\lambda }\left(k_{2}-k_{1}\right)L\cdot \log_{10}e$$

The coupling loss between the silicon waveguide and the GST waveguide is another contribution to the device loss. The power coupling rate (PCR) gives the total input coupling, taking into account both the modal overlap and the mismatch in effective indices between two modes. In simple cases, PCR is the product of the modal overlap integral (OI) and the Fresnel transmission rate (FR), written as [19,20]: (5)PCR=OI·FR
the overlap integral gives the fractional power coupling from the silicon waveguide ($\vec{E_{1}}$, $\vec{H_{1}}$) into GST waveguide ($\vec{E_{2}}$, $\vec{H_{2}}$), given by [19].
(6)OI=Re·[(∫E→1×H→2*·dS→)(∫E→2×H→1*⋅dS→)∫E1→×H1→⋅dS→]·1Re·(∫E2→×H2→⋅dS→)
the Fresnel transmission rate is given by
(7)$$\mathrm{FR}=\;\frac{2n_{Si}\;\cdot {\;\cos\;i}_{1}}{n_{Si}\;\cdot {\;\cos\;i}_{1}+n_{GST}\;\cdot {\;\cos i}_{2}}$$
where i_1_ and i_2_ represent the incident and refraction angles, respectively.

## 3. Design and Simulations

We first studied the GST waveguide using the finite-difference-time-domain (FDTD) simulations. The under-cladding of the device is a 2-μm-thick SiO_2_ layer on the silicon substrate and the upper-cladding is air. The complex refractive indices of GST around the 1550 nm wavelength are taken as 3.98 + 0.024i and 6.49 + 1.05i for amorphous and crystalline states, respectively [9]. Figure 2a,c show the forward optical power transmission along the x-axis (P_x_) in the amorphous GST waveguide when the GST waveguide width is 0.4 μm and 0.6 μm, respectively. Figure 2b,d show the P_x_ power distributions in their corresponding silicon waveguides, when their effective indices are matched with their amorphous GST waveguides. The heights of the GST and silicon waveguides are chosen as 0.05 μm and 0.22 μm. It can be seen that the optical power is mainly distributed outside the GST region for the 0.4-μm-wide GST waveguide, which has a high mode mismatch with the silicon waveguide. Therefore, we avoid the low-confinement GST waveguide in device optimization. 

We next scanned the GST waveguide width and height in order to get an optimal design. Figure 3a,b show the propagation loss of the GST waveguide at the amorphous and crystalline states, respectively. The amorphous GST has a lower propagation loss. For a thin GST layer (less than ~75 nm), the loss decreases for a narrower GST waveguide. This is because the light cannot be well confined in the GST layer when the size is small, leading to an expanded mode. When the GST layer is thick enough (larger than ~75 nm), the loss is first reduced and then almost unchanged with an increasing waveguide width. On the other hand, the crystalline GST has good confinement of light even with a small waveguide size. It has a much larger propagation loss than the amorphous state. For a fixed width, the loss increases with the height and then slightly decreases; for a fixed height, the loss always decreases with a wider waveguide. Figure 3c shows the ER as a function of GST height with GST width as a variable. It generally follows the same trend as the waveguide propagation loss at the crystalline state. 

To get a balanced performance between the insertion loss and propagation extinction ratio, we chose W_gst_ = 0.45 μm and H_gst_ = 0.12 μm. The GST waveguide effective index is 2.292 + 0.0247i at the amorphous state. The silicon waveguide dimensions are chosen to be 0.45 μm (width) × 0.22 μm (height). The modal effective index is calculated to be 2.29, which is close to the effective index of the amorphous GST waveguide. Figure 4a,b show the mode profiles for the silicon and GST waveguides, respectively. Although their refractive indices are close, the mode profiles are still different, resulting in mode transition loss at the waveguide interface. The calculated PCR is 0.911. Figure 4c,d show the light propagation at the two GST states. The GST section length is L_GST_ = 0.6 μm. The transmission losses at the amorphous and crystalline states are 0.70 dB and 21.04 dB, respectively. 

There is a compromise between the device IL and the ON-OFF switching ER. The IL is defined as the device optical transmission loss at the amorphous state. The ER is defined as the transmission loss difference between the amorphous state and the crystalline state. Figure 5 shows the optical transmission losses at the two states and the ER as a function of wavelength and GST length. The loss of ON-state (amorphous GST) increases when the wavelength deviates from the targeted 1550 nm wavelength because of the effective index mismatch. The loss of OFF-state (crystalline GST) does not monotonically increase with the GST length due to the Fabry-Perot resonant effect, given that the waveguide interface causes certain back reflection. It can be seen that when L_GST_ is 0.6 μm, the device IL is less than 1 dB, and the ON-OFF switching ER is more than 20 dB in an optical bandwidth of 70 nm. When the GST section is longer than 1 μm, the ER can be further improved but at the cost of a higher IL.

In the above device, the slight difference in modal profiles results in extra scattering and reflection losses in the waveguide junction. To mitigate this issue, we revised the design, as shown in Figure 6a. The GST waveguide dimensions are the same as the previous design. The silicon waveguide has two layers of tapers with heights of 70 nm and 150 nm. The height values comply with the silicon etch depths in silicon photonics foundries [10] so that it can be conveniently integrated with other optical devices. The width of the top taper is reduced from 500 nm to 80 nm, causing the light field to be gradually squeezed to the bottom taper. To match the mode profile of the GST waveguide, the bottom taper end width is chosen to be 700 nm. The length of upper taper (L_1_) and lower taper (L_2_) are determined to be L_1_ = 1.9 μm and L_2_ = 2 μm. The transition loss through the two tapers is less than 0.1 dB.

Figure 6b,c, show that the mode profiles of the silicon and GST waveguide match better than the previous design. The PCR is close to 0.968. Light propagates through the GST section with higher transitivity in the amorphous state but highly blocked in the crystalline state, as illustrated in Figure 6d,e.

Figure 7 shows the loss at the amorphous and crystalline states and the ER of the revised design. When L_GST_ = 0.7 μm, the device IL is 0.48 dB and the ER is as high as 33.1 dB at the wavelength of 1550 nm. In an optical bandwidth of 60 nm, the IL is less than 0.5 dB and the ER is larger than 30 dB.

## 4. Fabrication Tolerances

The proposed structures can be readily fabricated using complementary metal-oxide-semiconductor (CMOS) compatible fabrication processes. To investigate the robustness of our memristive switch to the alignment errors between the GST and Si waveguide layers, we simulated device performances (IL and ER) when the lateral alignment error increases. Table 1 shows the results. The operation wavelength is 1550 nm. The GST length is 0.7 μm. It can be seen that the IL increases and ER decreases when the misalignment rises from 0 to 20 nm. Nonetheless, the IL is still within 1 dB and the ER is larger than 30 dB for an alignment error up to 20 nm. It indicates that our device has a high tolerance to fabrication uncertainties.

## 5. Conclusions

GST is a promising material for optical switches due to its non-volatile high refractive index contrast between its amorphous and crystalline states. We have proposed an ultra-compact optical switch based on the phase change of GST to control the transmittance. We optimized the device to obtain a high ER and a low IL. The device with a 0.7-μm-long GST section can offer a high ER of more than 30 dB and a low IL of less than 0.5 dB with a broad optical bandwidth of 60 nm. The memristive switch shows high tolerance to misalignment between the GST and silicon waveguides. When the lateral alignment drift is up to 20 nm, the IL remains less than 1dB and the ER is larger than 30 dB. This ultra-small memristive switch can be applied in inter- and intra-chip optical communications.

## Figures and Tables

**Figure 1 micromachines-10-00453-f001:**
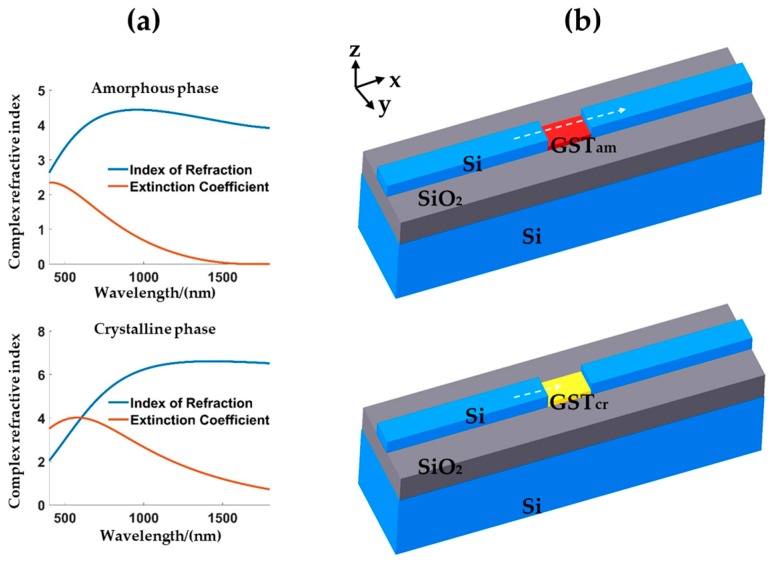
(**a**) Complex refractive index as a function of wavelength for amorphous and crystalline phases of GST material. (**b**) Schematic structure of the device showing ON-state and OFF-state optical transmissions.

**Figure 2 micromachines-10-00453-f002:**
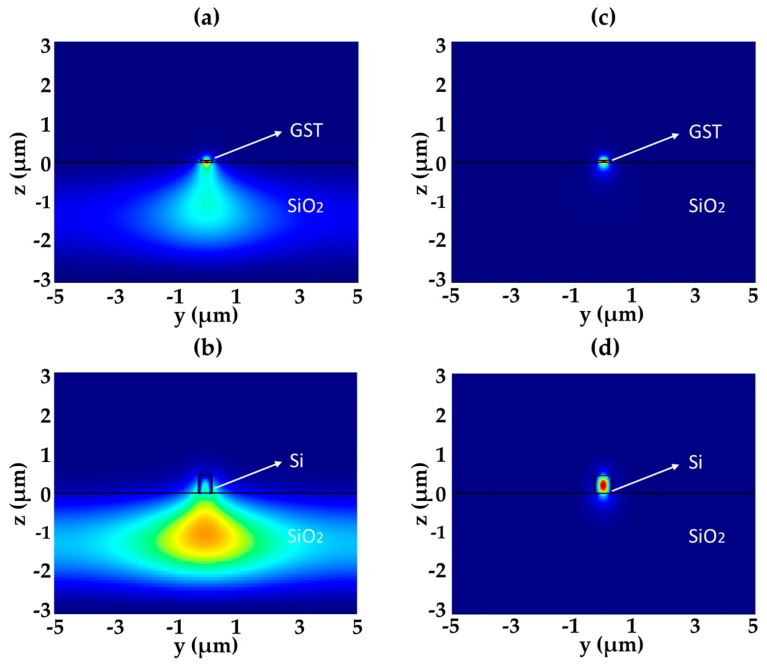
Cross-sectional P_x_ distributions in the (**a**) 0.4-μm-wide GST waveguide and (**b**) its corresponding silicon waveguide. Cross-sectional P_x_ distributions in the (**c**) 0.6-μm-wide GST waveguide and (**d**) its corresponding silicon waveguide. The GST layer thickness is 0.05 μm and the silicon waveguide layer thickness is 0.22 μm.

**Figure 3 micromachines-10-00453-f003:**
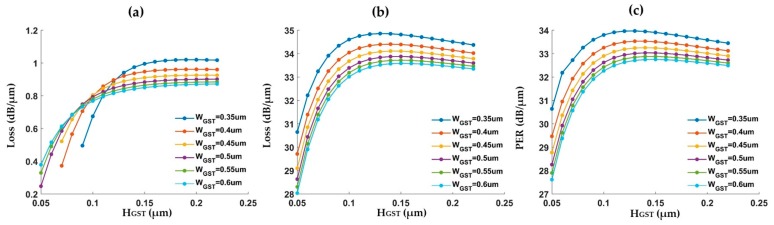
(**a**) Propagation loss of GST waveguide at the amorphous state. (**b**) Propagation loss of GST waveguide at the crystalline state. (**c**) Extinction ratio between the two states.

**Figure 4 micromachines-10-00453-f004:**
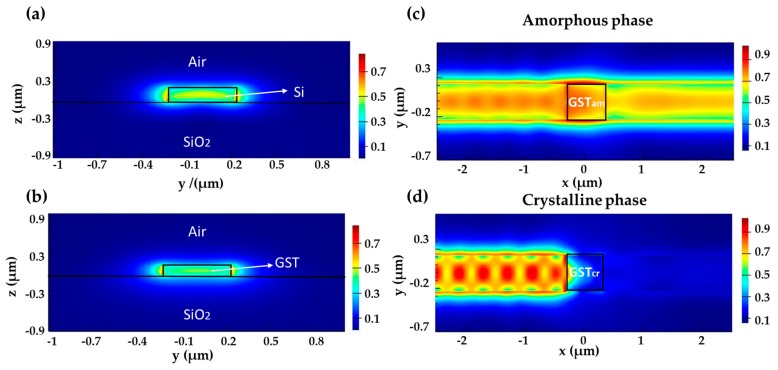
(**a**,**b**) Electric-field intensity mode profiles of (**a**) silicon waveguide and (**b**) amorphous GST waveguide. (**c**,**d**) Electric-field intensity distribution in the horizontal plane along the waveguide when GST is at (**c**) the amorphous state and (**d**) the crystalline state.

**Figure 5 micromachines-10-00453-f005:**
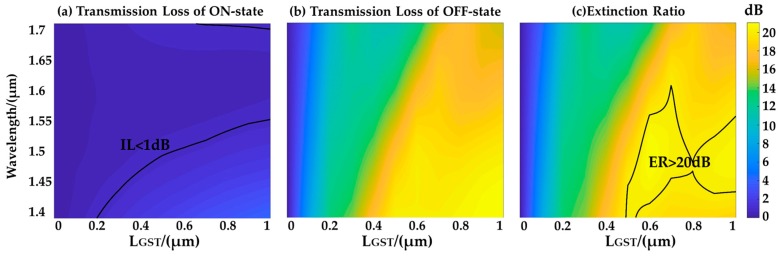
(**a**) Transmission loss of ON-state (amorphous phase). (**b**) Transmission loss of OFF-state (crystalline phase). (**c**) ON-OFF switching extinction ratio after phase change of GST.

**Figure 6 micromachines-10-00453-f006:**
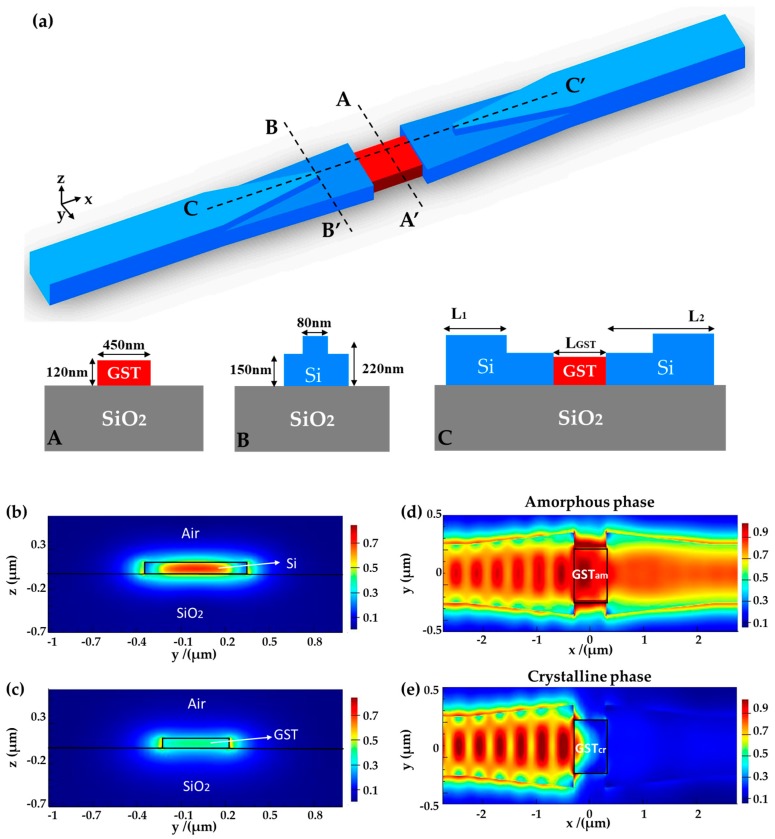
(**a**) Three-dimensional view of the revised design. The insets show cross-sectional views along the segments AA’, BB’, and CC’. (**b**,**c**) Electric-field intensity profiles of (**b**) the silicon waveguide and (**c**) the amorphous GST waveguide across the interface. (**d**,**e**) Electric-field intensity distributions in the horizontal plane along the waveguide when GST is at (**d**) the amorphous and (**e**) the crystalline states.

**Figure 7 micromachines-10-00453-f007:**
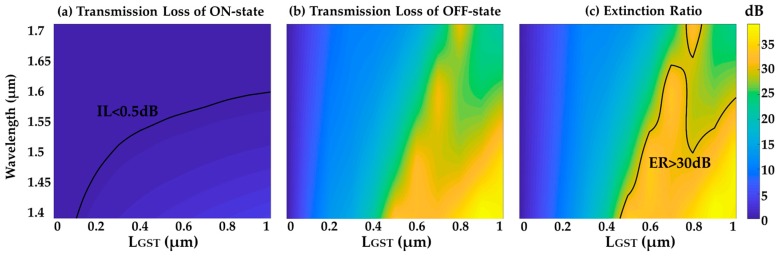
(**a**) Transmission loss of ON-state (amorphous phase). (**b**) Transmission loss of OFF-state (crystalline phase). (**c**) ON-OFF switching extinction ratio when GST phase change occurs.

**Table 1 micromachines-10-00453-t001:** ON-state insertion loss and ON-OFF switching extinction ratio for different lateral alignment errors between the GST and Si waveguides.

Misalignment	IL (Insertion Loss)	ER (Extinction Ratio)
0 nm	0.48 dB	33.1 dB
5 nm	0.79 dB	32.6 dB
10 nm	0.80 dB	32.1 dB
15 nm	0.81 dB	31.7 dB
20 nm	0.82 dB	31.2 dB

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
