# Peer review of "Design of Ultra-Compact Optical Memristive Switches with GST as the Active Material"

_micromachines, 2019, doi:10.3390/mi10070453_

Round 1

Reviewer 1 Report

The article reports on a phase change switch based on GST material. Such switches have been intensively studied based on the large permanent change induced in the material making them attractive over thermo- or electro-optical switches. The article is a theoretical study that is well laid out, but the authors should address the following:

-      How does this result compare in terms of performance to many other recent demonstrations? It is hard to determine the advance over prior work without some performance comparison.

-      How was the refractive information in figure 1a measured? This information should be included.

-      All variables should be defined (e.g. i1 and i2 in equation 7)

-      What is the size of the silicon strip waveguide?

-      It is not clear how the choice between 0.4-um and 0.6-um-wide GST waveguides arises. It seems arbitrary. Why were these widths initially explored? Please add explanation. Also, in Figure 2, it would be helpful to see both the silicon waveguide mode profile and the GST mode profile for different GST dimensions.

-      Figure 2 caption: what is Px?

-      Figure 3 is missing a fourth plot: mode overlap loss (or PCR) between silicon and amorphous and crystalline GST waveguides.

-      Figures 4 and 6 are missing the mode profile for the crystalline GST waveguide

-      Can the authors comment on how the GST waveguide can be fabricated with a smooth transition in the middle of two silicon waveguides? Has this been shown to be feasible in demonstrated structures?

Author Response

Thank you for your valuable advice! Please see the attachment.

Reviewer 2 Report

It is a nice work with systematical and complete analysis to a memristive switch combing GST material and silicon waveguide. However, the design is too ideal to practically realize. I suggest the authors analyze the fabrication tolerance e.g. the influences of misalignment between GST and the Si waveguide on the device performances.

Author Response

(The authors gave the same response as above.)
